# Transcriptome Analysis Reveals the Inducing Effect of *Bacillus siamensis* on Disease Resistance in Postharvest Mango Fruit

**DOI:** 10.3390/foods11010107

**Published:** 2022-01-01

**Authors:** Zecheng Jiang, Rui Li, Yue Tang, Ziyu Cheng, Minjie Qian, Wen Li, Yuanzhi Shao

**Affiliations:** 1College of Food Science and Engineering, Hainan University, Haikou 570228, China; jiangzecheng516@163.com (Z.J.); ty15828556490@163.com (Y.T.); chengzi_yu@163.com (Z.C.); 2Key Laboratory for Quality Regulation of Tropical Horticultural Crops of Hainan Province, College of Horticulture, Hainan University, Haikou 570228, China; ruileemo@163.com (R.L.); cookiekin@zju.edu.cn (M.Q.); 3School of Life Sciences, Hainan University, Haikou 570228, China

**Keywords:** *Bacillus siamensis*, mango fruit, disease resistance, transcriptome analysis, gene expression

## Abstract

Postharvest anthracnose, caused by the fungus *Colletotrichum gloeosporioides*, is one of the most important postharvest diseases of mangoes worldwide. *Bacillus siamensis* (*B. siamensis*), as a biocontrol bacteria, has significant effects on inhibiting disease and improving the quality of fruits and vegetables. In this study, pre-storage application of *B. siamensis* significantly induced disease resistance and decreased disease index (DI) of stored mango fruit. To investigate the induction mechanisms of *B. siamensis*, comparative transcriptome analysis of mango fruit samples during the storage were established. In total, 234,808 unique transcripts were assembled and 56,704 differentially expressed genes (DEGs) were identified by comparative transcriptome analysis. Gene ontology (GO) enrichment and Kyoto Encyclopedia of Genes and Genomes (KEGG) pathway analysis of DEGs showed that most of the DEGs involved in plant-pathogen interaction, plant hormone signal transduction, and biosynthesis of resistant substances were enriched. Fourteen DEGs related to disease-resistance were validated by qRT-PCR, which well corresponded to the FPKM value obtained from the transcriptome data. These results indicate that *B. siamensis* treatment may act to induce disease resistance of mango fruit by affecting multiple pathways. These findings not only reveal the transcriptional regulatory mechanisms that govern postharvest disease, but also develop a biological strategy to maintain quality of post-harvest mango fruit.

## 1. Introduction

Mango fruit (*Magnifera indica* L.) is one of the most popular fruits with delicious taste, rich nutrition, and a variety of bioactive compounds [1]. However, mango fruit is highly perishable and susceptible to various pathogenic fungi during postharvest. Anthracnose, caused by *Colletotrichum gloeosporioides*, can cause losses of up to 30 percent the during storage and transportation of mango fruit [2]. Therefore, control of the postharvest disease of mango fruit is still a major issue for decreasing postharvest losses and accelerating the development of the mango industry. Presently, many control strategies against postharvest diseases have been developed. Some physical technologies, such as postharvest heat treatment and UV-C treatment, have been found to effectively control the postharvest diseases of fruits [3]. Synthetic fungicides including mancozeb, carbendazim, promethanide, and tecto60 have been confirmed to have obvious effects on inhibiting anthracnose in mango fruit [4]. Although these fungicides are effective for controlling pathogens, they do have negative influences on human health and the environment [5]. Therefore, seeking effective, safe, and valuable alternative fungicides to control postharvest anthracnose disease in mango is of great significance.

As a biological control strategy, microbial antagonists possess a great potential to fight against phytopathogens and have developed into a promising alternative in postharvest disease management [6]. *Bacillus*, a genus of bacteria, can form spores (endospores), have strong resistance to external harmful factors, and are widely distributed in soil, water, air, and plants. *Bacillus*
*siamensis*
*(B.*
*siamensis**)*, as a biocontrol antagonist, was observed to have significant effects on inhibiting disease, improving the quality of fruits and leafy vegetables [7]. *B. siamensis* was also confirmed to have biocontrol efficacy against brown spot disease of tobacco [8], and the induced tolerance by *B. siamensis* to salinity stress was found in wheat [9]. According to the present research, the main mechanism of biological antagonism could be contributed to multiple processes, including producing bacteriostatic substances, competing for nutrition and space with pathogens, secreting extracellular lyase, and so on [10,11].

In recent years, high-throughput sequencing technology (RNA-seq) has developed rapidly, and it could be used to provide a comprehensive analysis of the genome and transcriptome of species and reveal the gene expression of individual organisms in specific developmental stages and specific tissues [12]. RNA-seq has become a tool for studying the molecular mechanisms of regulating various physio-chemistry processes, including fruit quality changes, disease response, and so on [13]. Zhao et al. [14] reported the resistance mechanism of citrus by transcriptome sequencing of Geotrichum citri-auranti. Xu et al. [15] explained the complex defense response induced by Talaromyces rugulosus O1 infection in grapes by RNA-seq analysis. The results of these studies suggest that many biological pathways, including plant hormones, plant-pathogen interactions, reactive oxygen species (ROS), biological stimuli, hormone signaling, and a range of resistance genes, are involved in plant defense systems.

Previously, we have evaluated the effects of *B. siamensis* against pathogens of tropical fruits, such as mango and lychee, and explored the possible physiological mechanisms in vivo and in vitro [16]. However, the molecular mechanisms of *B. siamensis* inducing disease resistance in mango fruit have rarely been investigated. The profiles of transcriptomic and gene expression induced by *B. siamensis* may be crucial for the enhancement of defense response in mango fruits. Therefore, exploring and understanding the induction mechanisms mediated by *B. siamensis* in harvested fruits is beneficial for us to improve the controlling disease technique. In this study, the effect of *B. siamensis* treatment on controlling decay index caused by anthracnose in mango fruit stored at 25 °C was determined. The differential genes and regulatory pathways that related to the disease resistance of mango fruit were identified through transcriptomic analysis. Moreover, the patterns and changes in the expression of key differential genes related to the disease response of mango fruit were investigated. 

## 2. Materials and Methods

### 2.1. Strain

*Bacillus siamensis* (*B. siamensis*) strain was isolated and identified in our laboratory in 2019. Firstly, appropriate *B. siamensis* was taken from the inclined plane and cultured in Nutrient Agar (NA) at 28 °C for 48 h. Then, *B. siamensis* on NA was extracted into 250 mL Nutrient Broth (NB). It was cultured at 180 rpm at 28 °C for 20 h. The number of spores was measured with a blood cell counting plate and the concentration of bacterial suspension was adjusted to 1 × 10^8^ cells/mL for the next processing. 

### 2.2. Plant Materials

Commercial mature mango cultivar ‘Tainong’ fruits were harvested from a commercial orchard in Lingshui County, Hainan province, and transported to the laboratory as soon as possible. Three hundred uniform fruits without physical injury were chosen and randomly divided into two groups for soaking with tap water (marked with control) and *B. siamensis* suspension (10^8^ cells/mL, marked with treatment) for 20 min. Polyethylene film bag (0.02 mm in thickness, Fuxiang, Shenzhen) wrapped fruits were stored for 9 days at the temperature and relative humidity of 25 °C and 95%, respectively. Of the 150 fruits, 30 were labeled for the observation of disease index (DI) and the rest for sampling. Mixed samples of peel and flesh were collected at 0, 3, 6, and 9 days after treatment, with one randomly chosen fruit per biological replicate. All treatments were conducted with three biological replicates. 

### 2.3. Evaluation of Biocontrol Effect of B. siamensis on Pathogens in Stored Mango Fruit

According to the proportion of the diseased spot area to the fruit surface area, the mangoes in each group were classified as grade 0 (no diseased spot), grade 1 (diseased spot area <1/10), grade 2 (diseased spot area accounted for 1/10~1/5), grade 3 (diseased spot area accounted for 1/5~1/2), and grade 4 (diseased spot area >1/2). The disease index (DI) was calculated with following formula: DI = ∑ (number of fruits in each disease grade × the disease grade)/(total number of investigations × the highest disease grade). 

### 2.4. Construction and Sequencing of the Transcriptome 

In the present study, twenty-one RNA-Seq libraries for three biological replicates of each of the four time points (0 d, 3 d, 6 d, and 9 d) of the two groups (control and *B. siamensis*) were constructed. Total RNA was extracted using the Trizol reagent kit (Invitrogen, Carlsbad, CA, USA) according to the manufacturer’s protocol. RNA quality was assessed on an Agilent 2100 Bioanalyzer (Agilent Technologies, Palo Alto, CA, USA) and checked using RNase free agarose gel electrophoresis. 

mRNA was enriched, fragmented, and reverse transcribed into cDNA with random primers. After the second-strand cDNA was synthesized, cDNA fragments were purified with the QiaQuick PCR extraction kit (Qiagen, Venlo, The Netherlands), end repaired, A base added, and ligated to Illumina sequencing adapters. After agarose gel electrophoresis, suitable ligation products were selected as templates for PCR amplification, and the library was sequenced using Illumina novaseq 6000 by Gene Denovo Biotechnology Co. (Guangzhou, China). Three biological replicates were performed.

Fastp (version 0.18.0, Shenzhen, China) was used to further filter out the raw reads containing adapters or low quality bases to obtain a high quality clean read. Then, the unigene expression was calculated and normalized to FPKM (fragment per kilobase of transcript per million mapped reads). 

#### 2.4.1. Unigene Functional Annotation

BLAST alignment was used to annotate the assembled unigenes. The public databased used for BLAST included the NCBI non-redundant protein database (NR, http://www.ncbi.nlm.nih.gov, accessed on 20 August 2021), Swiss-Prot (http://www.expasy.ch/sprot, accessed on 20 August 2021), COG/KOG (http://www.ncbi.nlm.nih.gov/COG, 20 August 2021), Gene Ontology (GO, http://www.geneontology.org, accessed on 20 August 2021), and the Kyoto Encyclopedia of Genes and Genomes (KEGG, http://www.genome.jp/kegg, accessed on 20 August 2021) database.

#### 2.4.2. Differentially Expressed Genes (DEG) Analysis

RNAs differential expression analysis was performed by DESeq2 software between two different groups (and by edgeR between two samples). The genes with the parameter of false discovery rate (FDR) below 0.05 and absolute fold change ≥2 were considered differentially expressed genes.

#### 2.4.3. GO Enrichment and Pathway Analysis of DEG

An online platform named OmicShare tools (http://www.omicshare.com/tools, accessed on 15 September 2021) was used to obtain the Gene Ontology (GO) annotation and KEGG annotation of DEGs (CK-3d-vs-BS-3d, CK-6d-vs-BS-6d, and CK-9d-vs-BS-9d). GO terms and KEGG terms with corrected *p* value ≤ 0.05 were considered significantly enriched.

### 2.5. The Quantitative Real-Time PCR (qRT-PCR) Analysis 

DEGs related to fruit resistance against pathogen infection were screened out, and the selected genes were verified by quantitative reverse transcription polymerase chain reaction (qRT-PCR). Total RNA was extracted by the CTAB method [17]. First-strand cDNA was synthesized from 1 μg RNA using the HiScript II Q RT SuperMix (Novizin Bio, Nanjing, China). The primers were designed using the Primer explorer v5 online website (https://primerexplorer.jp/e/, accessed on 10 December 2020) and are listed in Table 1. The ACTIN gene was used as a reference gene. For each sample, three replicates were performed at a final volume of 10μL, which consisted of 3.2 μL of ddH2O, 0.8 μL of primers, 1 μL of cDNA, and 5 μL of 2 × Q3 SYBR qPCR Master mix (Tolo Biotech, Shanghai, China). The thermal cycling protocol was 5 min at 95 °C, and then 40 cycles of 95 °C for 5 s and 30 s at 60 °C for annealing and extension using a qTOWER3 G Real-Time PCR System (Wacker Biotech GmbH, Jena, Germany). The 2^−ΔΔCT^ method was used to calculate the relative expression of genes in the samples. 

### 2.6. Statistical Analysis

The data were analyzed by analysis of variance (ANOVA) using the statistical program SPSS/PC version II.x (SPSS Inc. Chicago, IL, USA), and the Duncan’s multiple range test was used for mean separation. The statistical significance was assessed at *p* ≤ 0.05. Origin 2018 (Origin 2018, Hampton, MA, USA) was used for graphs construction.

## 3. Results

### 3.1. The Effect of B. siamensis on Disease Index (DI) of Mango Fruit Stored at 25 °C 

As shown in Figure 1A, *B. siamensis* treatment significantly slowed down the anthracnose symptoms of mango fruit during the storage. The DI increased during the whole storage period both in the control and in treated fruits (Figure 1B); however, the increase speeds were markedly different between the control and treated fruits, as the DI of *B. siamensis* treated fruits were lower 0.1 and 0.24 than those of control fruit on 6 d and 9 d, respectively (*p* ≤ 0.05). This indicates that the disease occurrence of mango fruit could be effectively inhibited by *B. siamensis* treatment.

### 3.2. Sequencing and Transcriptome Assembly

A total of 850,798,964 raw reads were obtained in mango fruit by Illumina HiseqTM 2500 paired-end sequencing. After a stringent filtering process, 839,584,440 higher quality clean reads with 38.9311% of GC remained (Table 2). 

Based on the clean reads, a total of 56,704 unigenes were assembled by the Trinity program. A total of 35,648 (62.87% of all unigenes) unigenes were annotated (Table 2).

### 3.3. Statistics of Different Expressed Genes (DEGs)

The variation of differential genes in all groups after comparison was shown in Figure 2A. Compared with day 0, there were more and more DEGs in both the treatment group and control group, and there were fewer up-regulated genes than down-regulated ones. For the control fruit, 1890 genes of up-regulation and 4262 genes of down-regulation were observed from day 0 to day 3 (C0d-vs-CK-3d). In groups of C0d-vs-CK-6d and C0d-vs-CK-9d, there were much more decreased genes than up-regulated genes. Compared with the control fruit, 2302 genes were up-regulated by *B. siamensis* treatment on day 3, and expressions of 4117 and 4681 genes were increased by treatment on day 6 and day 9, respectively. 

The expression abundance of genes in all samples is presented in Figure 2B. Compared with day 0, the expression abundance of genes decreased with the extension of storage time. There was no significant difference in overall gene expression between the treatment and control. Because the violin diagram shows the shape of large in the middle and small at the two ends, it indicates that the expression level distribution of day 0 was less uniform than that of other treatment groups. On the third day, both treatment and control expression levels were evenly distributed. The expression levels of all groups had obvious discrete values.

### 3.4. GO and KEGG Enrichment Analysis of DEGs in Mango Fruit

The DEGs of the three comparison groups were collected and then analyzed by GO and KEGG analysis. As shown in Figure 3A, DEGs in the three comparison groups of mangoes can be divided into biological processes, molecular functions, and cellular components, and were further divided into 45 secondary functional groups. Among those groups, 23 subclasses were induced by biological processes. Metabolic process, cell process, and single tissue process were the most abundant of the biological processes. Molecular function includes 11 subclasses, among which catalytic activity and binding activity are the most enriched. Cellular components include 11 subclasses. For cellular components, the organelles were the most abundant among them. 

The 20 most significantly enriched pathway entries were selected and are shown in Figure 3B. The first three pathways are metabolic pathways, secondary metabolites, and starch and sucrose metabolism. Six pathways that are associated with disease resistance are highlighted in the orange box; they were phenylpropanoid biosynthesis, plant hormone signal transduction, flavonoid biosynthesis, plant-pathogen interaction, stilbenoid, diarylheptanoid and gingerol biosynthesis, and peroxisome pathway. 

### 3.5. Analysis of Differential Expression Genes (DEGs)

For comparing the expression difference of key genes between the control and treatment, the following three groups were selected from the total comparison group (Figure 4A): CK-3d-vs-BS-3d (up: 802, down: 931), CK-6d-vs-BS-6d (up: 60, down: 173), and CK-9d-vs-BS-9d (up: 515, down: 164). The number of DEGs in different groups was calculated in the histogram (Figure 4A). The CK-3d-vs-BS-3d group had more DEG than the other two groups. 

Venn diagrams of DEGs in the three comparison groups were generated to show the overlap between the number of DEGs and the comparison groups (Figure 4B). There were 10 overlapping DEGs in all three groups. The CK-3d-vs-BS-3d group and CK-6d-vs-BS-6d group had 50 overlapping DEGs, and there were 131 DEGs overlapped with the CK-9d-vs-BS-9d group. CK-6d-vs-BS-6d and CK-9d-vs-BS-9d were overlapped by 25 DEGs. 

In the different controls, the volcano plot visually showed the relationship between processing and up-regulation/down-regulation of DEGs number (Figure 4C). We found that in the CK-3d-vs-BS-3d group, the number of DEGs that were up-regulated and down-regulated were basically the same. DEGs were mainly up-regulated in the CK-9d-vs-BS-9d comparison group. 

### 3.6. The Key Genes Analysis in Pathways Related with Disease-Resistance of Mango Fruits 

Six pathways related to the disease-resistance of mango fruits are listed in Appendix A. The key genes selected in six pathways are exhibited in the Heat map (Figure 5). In plant hormone signal transduction pathway, four genes (*PR1, JAZ, BAK1,* and *GH3*) were up-regulated and the *NPR1* gene in fruits was down-regulated by *B. siamensis* treatment. In the plant-pathogen interaction pathway, the HSP90A gene was up-regulated and eight genes (*WRKY1, WRKY2, WRKY22, PTI5, PTI6, CNGC, FLS2,* and *MPK3*) were down-regulated after 0 days. In the peroxisome pathway, the SOD gene was up-regulated and the *CAT* gene was down-regulated by *B. siamensis* treatment. In the pathway of phenylpropanoid biosynthesis, *B. siamensis* treatment enhanced the expression of the *PAL* gene and decreased the expression of the *4CL* gene. In flavonoid biosynthesis, two genes (*CHS, E5.5.1.6*) were down-regulated after 0 days. In the gingerol biosynthesis pathway, *HCT* and *CYP73A* were up-regulated by *B. siamensis* treatment at 9 days. 

### 3.7. Validation of Gene Expression Profiles by qRT-PCR

In order to validate gene expression profiles, 14 unigenes, including genes of *PR1, NPR1, JAZ,* and *GH3*, which related to plant hormone signal transduction pathway; genes of *WRKY2, WRKY22, PTI6,* and *MPK3*, which related to plant-pathogen interaction pathway; genes of *IDH1, CAT, SOD1,* and *SOD2*, which related to peroxisome pathway; and genes of *PAL* and *4CL*, which related to phenylpropanoid biosynthesis pathway, were selected for qRT-PCR analysis. 

The expression levels of these key genes are shown in Figure 6. In the pathway of plant hormone signal transduction, the expression levels of *PR1* and *BAK1* were greatly increased by *B. siamensis* treatment during the whole storage. The expressions of *GH3* in treated fruit were distinctly higher than those in the control on days 6 and 9, while expression of *NPR1* in treated fruit was higher than that in the control only on day 3. 

In the plant-pathogen interaction pathway, expressions of *WRKY2, WRKY22, PTI6,* and *MPK3* in the treated fruit were higher than those in the control group at 3, 6, and 9 days. For the peroxisome pathway, the significant higher expressions of *IDH1, CAT, SOD1,* and *SOD2* were observed in treated fruits than those of the control fruit during storage of 3 days. In the phenylpropanoid biosynthesis pathway, the expression of the *PAL* gene in treated fruit was significantly higher than that in the control fruit only on day 9. However, *4CL* gene expressions in treatment group were significantly higher than those in the control group during the whole storage. 

FPKM values from transcriptome data are also presented in Appendix A. Correlation analysis (Appendix A) showed that the expression levels of genes in *PTI6*, *MPK3,* and *IDH1* had significant correlations with their FPKM values (R^2^ ≥ 0.99); they were followed by genes of *SOD1, NPR1,* and *SOD2* (R^2^ ≥ 0.97). However, no significant relationship between gene expression and FPKM was observed in the *WRKY2* gene (R^2^ = 0.534).

## 4. Discussion

Anthracnose decay is one of the most common and vital fungal diseases of mangoes, and the parameters of DI are regarded as indicators of disease severity in mangoes [18]. In this study, *B. siamensis* treatment significantly alleviated anthracnose symptoms and reduced the disease index (Figure 1), suggesting that *B. siamensis* treatment enhanced the disease-resistance of stored mango fruits and again confirmed the findings reported in our former studies [16]. Postharvest *B. siamensis* application also suppressed the development of DI in chickpea and raspberries [19,20]. The results suggest that *B. siamensis* could be used as a biological technique to reduce postharvest diseases of fruit and vegetables. Currently, RNA-seq analysis has been employed to study the interactions between antagonistic microorganisms and fruit, and some impressive results have been achieved [21]. Up to now, few studies and reports are available on the molecular response of the mango fruit to *B. siamensis*.

In our study, compared with control, nearly 2917 genes were significantly affected by *B. siamensis*, 802 genes were up-regulated by *B. siamensis* treatment on day 3, and expressions of 60 and 515 genes were increased by treatment on day 6 and day 9, respectively (Figure 2A). There are more DEGs in mango fruit before 3 days of storage than those of 6 days and 9 days, indicating that the response of the fruit to *B. siamensis* started in the early storage days. There are more up-regulation DEGs than down-regulation DEGs on day 9 (Figure 4A), suggesting that the induced effects of *B. siamensis* on disease-resistance of fruit could be maintained until the end of storage. In this way, analysis of the possible functions of the DEGs from the comparison between them would help reveal the mechanisms of induced resistance.

Many previous studies demonstrated that genes related to L-phenylalanine metabolism, plant hormone signal transduction, programmed cell death (PCD) regulation, and biosynthesis of amino acids could be induced by antagonistic microorganisms [22,23]. In this study, we found that plant-pathogen interaction, plant hormone signal transduction, and biosynthesis of phenylpropanoid, flavonoid biosynthesis, stilbenoid, diarylheptanoid, and gingerol were the most enriched pathways, whether for control or treatment (Figure 3A,B), indicating that these processes were involved in mango response to *B. siamensis*.

Plant hormones and phytohormone signaling pathways regulate complex signaling networks associated with plant development and response to environmental stress [24]. An et al. [25] found that 17 genes were enriched in plant hormone signal transduction pathways after being treated with chitosan. *PR1* is considered to be a marker of enhanced defense status conferred by pathogen induced systemically acquired resistance (SAR). Yuan et al. [26] found that genes related to *PR1* were significantly enriched during fusarium graminearum infection. These findings indicate that the *PR1* gene plays an important role in the process of plant and pathogen interaction. Our present study showed that the *PR1* gene was significantly up-regulated by *B. siamensis* treatment (Figure 5). Therefore, the increase in anti-disease ability of mango fruit pathogens could be contributed to the induced gene expression by *B. siamensis* treatment during the storage.

The role of jasmonic acid (JA) in disease resistance of fruit has been widely studied and reported; JA could be rapidly induced to produce an effective defense mechanism when plants are attacked by insects and fungi [27]. As a key regulator of the JA signaling pathway, the *JAZ* protein plays an important role in plant growth, development, and stress [28]. The *JAZ* gene is one of the genes encoding the *JAZ*s protein in the TIFY family. The *NPR1* gene is another key regulator of salicylic acid (SA) signaling, and it plays an important role in plant defense against pathogens [29]. In this study, the higher expression level of *JAZ* and the lower expression level of *NPR1* in *B. siamensis* treated fruit than those of control fruit were detected (Figure 5), suggesting that the *JAZ* gene plays a more important role than the *NPR1* gene in disease resistance of stored mango fruit. The *BAK1* gene, which regulates the plant immune response [30], was up-regulated by *B. siamensis* treatment in our study, indicating that the disease resistance of mango could be induced by *B. siamensis*.

In present study, a large number of DEGs in the pathway of plant-pathogen interaction were significantly influenced by *B. siamensis* treatment, suggesting that *B. siamensis* affected the complex signaling networks of stored mango fruit [31]. It has been reported that up-regulation of *MPK3* may improve plant resistance by regulating salicylic acid signal transduction [32]. As important modulators of immune response, many members of the plant-specific WRKY transcription factors are related to plant defense against pathogens. It has been found that the expression of *WRKY22* was up-regulated in the eliciting resistance of peach fruit to stolon nematodes [33]. In our study, expressions of *MPK3* and *WRKY22* genes in mango fruit were significantly up-regulated by *B. siamensis* treatment on day 3 of storage and, consequently, the anti-disease ability of fruit induced by *B. siamensis* was enhanced. 

In the peroxisome pathway, the accumulation of reactive oxygen species can systematically signal the induction of resistance genes. SOD and CAT are regarded as the enzymes that are most closely related to the metabolism of ROS, and they are dominant in reducing the ROS production, and regulating the balance between the ROS generating and scavenging system may be a way for plants to gain disease resistance [34]. In our study, *B. siamensis* treatment could reduce expression of *CAT*, while increasing the expression of *SOD*, and the increase in *SOD* activity could enhance the defense against pathogens [35]. 

In plants, phenylpropionic acid and flavonoid metabolic pathways are considered as two central defense signals under stress [36]. Phenylpropanoid compounds are important secondary metabolites, and these compounds can protect plant from stress damages due to their antioxidant attributes or other protective properties [37]. In the phenylpropanoid biosynthesis pathway, PAL is a key enzyme in the biosynthesis of lignin, salicylic acid, and other phenylalanine metabolism [38]. As a major component of plant cell walls, lignin contributes to the formation of physical and chemical barriers in plant immunity [39,40]. In study of pepper fruit, it was found that *PAL* related genes were activated earlier and changed more in the resistant population than in the susceptible population [41]. Because the phenylpropanoid biosynthesis is the starting point for biosynthesis of several metabolites, activating the phenylpropyl pathway can greatly enhance plant disease resistance [42]. Our present results showed that the expression of the *PAL* gene induced by *B. siamensis* increased continuously during the storage of mango fruit, while expression of the *PAL* gene in the control fruit remained stable. The results were consistent with the induction of the *PAL* gene in avocado fruit reported by Xoca-Orozco et al. [43]. The phenylpropane metabolic pathway is closely related to plant disease resistance. When plants are infected by pathogens, the expressions of *PAL* and other related genes can be induced, and the increasing in enzyme activity and accumulation of secondary metabolites can prevent the further infection of pathogens, which contribute to all aspects of plant responses towards biotic and abiotic stimuli [44]. 

Flavonoid pathway is a branch of phenylpropanine metabolism. Flavonoids provide protection by removing reactive oxygen species and limiting the cell death required by pathogens [45]. In the flavonoid pathway, 4CL is one of the key enzymes to affect the biosynthesis of flavonoids and aromatic compounds in plants. In the present study, it was found that the *4CL* gene in mango was significantly up-regulated under the induction of *B. siamensis*. *CHS* is another key gene in the flavonoid pathway, and it was also induced by *B. siamensis* in mango fruit in our study. Lei et al. [46] reported that the over-expression of *CHS* is beneficial to the accumulation of rutin and hesperetin. This suggests that the accumulation of flavonoid compounds can be increased by up-regulating flavonoid biosynthesis genes, thus improving the resistance of fruits against pathogens. 

Another important branch of the phenylpropionic acid defense pathway is the gingerol biosynthesis pathway. Studies have shown that ginger phenol is an important secondary metabolite, which can improve the immune response of plant [47]. In our study, *HCT* was up-regulated by *B. siamensis*, which was conducive to the accumulation of ferulic acid, the precursor of 6-gingerol, and was beneficial for the induction of disease resistance in stored mango fruit.

## 5. Conclusions

The summary of the mechanisms involved in the disease resistance of mango fruit induced by *B. siamensis* is highlighted in Figure 7, based on the results of transcriptome analysis. Firstly, some genes (*JAZ*, *BAK1*, and *PR1*) were up-regulated by *B. siamensis* treatment, which triggered a stress response, and induced synthesis of resistant substances of phenols, which improve the disease resistance of mango fruit. Secondly, some genes (*WRKY22*, *HSP90, CNGCs, SOD, PAL, 4CL, CHS,* and *HCT*) related to pathways of plant-pathogen interaction, such as peroxisome, phenylpropanoid, flavonoid, and gingerol biosynthesis, were up-regulated by *B. siamensis* in mango fruit, which enhance the system’s anti-disease ability, stimulate the immune response, and, finally, greatly enhance the disease resistance of mango fruit against pathogens. 

Overall, the disease resistance of mango fruit could be improved by *B. siamensis*. The results of this research provide a reference for future studies on controlling postharvest disease of fruit and vegetables using antagonistic bacteria and provide a theoretical basis for the practical application of an antagonistic agent.

## Figures and Tables

**Figure 1 foods-11-00107-f001:**
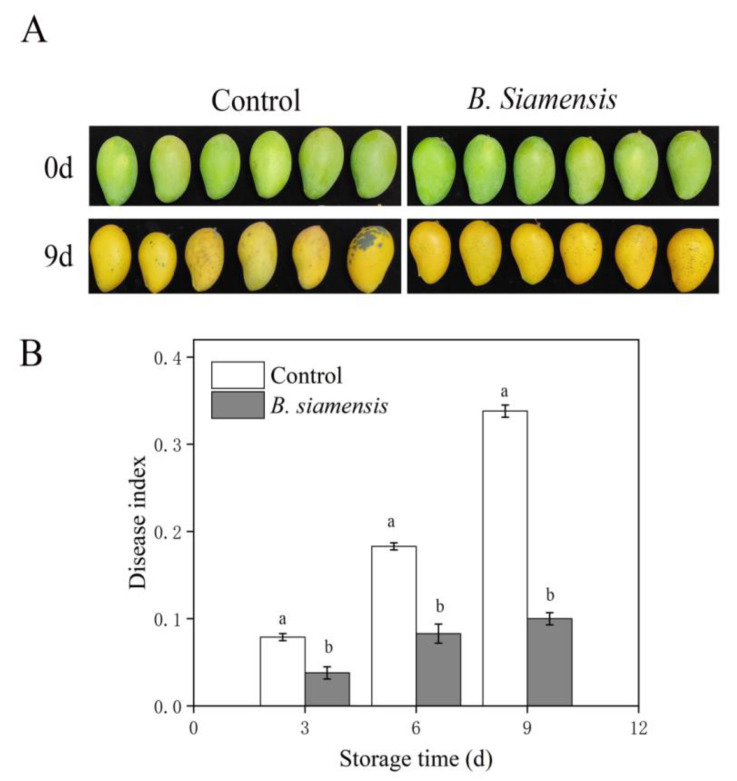
Effect of *B. siamensis* treatment on appearance and anthracnose symptoms (**A**) and disease index (**B**) of mango fruit stored at 25 °C for 9 days. Bars with different lowercase letters indicate significant differences based on a t-test at the *p* ≤ 0.05 level. Each value represents the mean ± SE of three replicates.

**Figure 2 foods-11-00107-f002:**
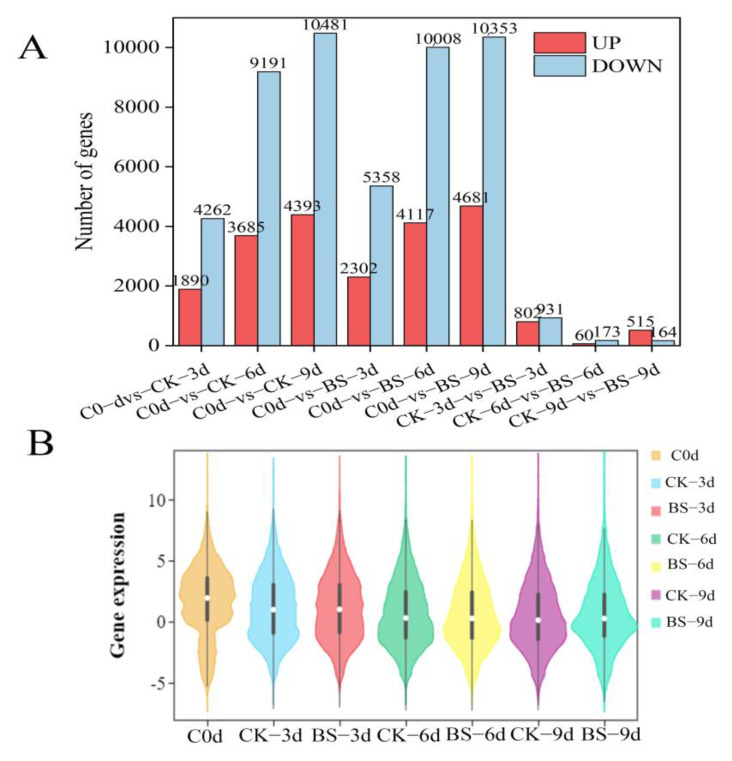
Statistics of differentially expressed genes in all mango fruit groups. (**A**) Bar chart of the number of all DEGs. (**B**) Violins in all groups (the white dot in the middle of the box is the median, the upper and lower edge of the box type is 75%, and the upper and lower limit is 90%).

**Figure 3 foods-11-00107-f003:**
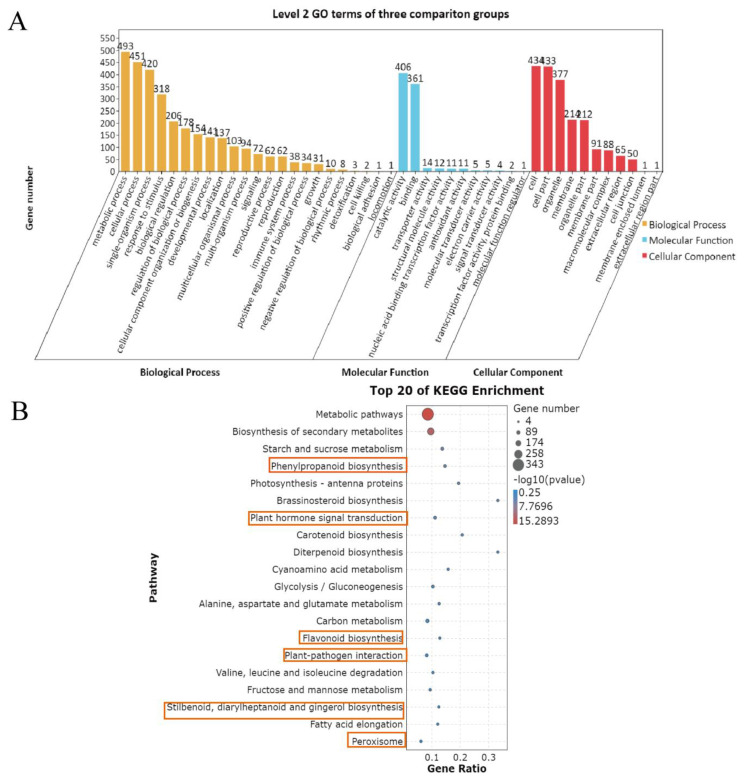
Transcriptomic profiles of mango fruit during storage at 25 °C. (**A**) GO functional classification of the DEGs. (**B**) Significant enrichment analysis of KEGG of the DEGs. Six pathways that are associated with disease resistance are highlighted in the orange box.

**Figure 4 foods-11-00107-f004:**
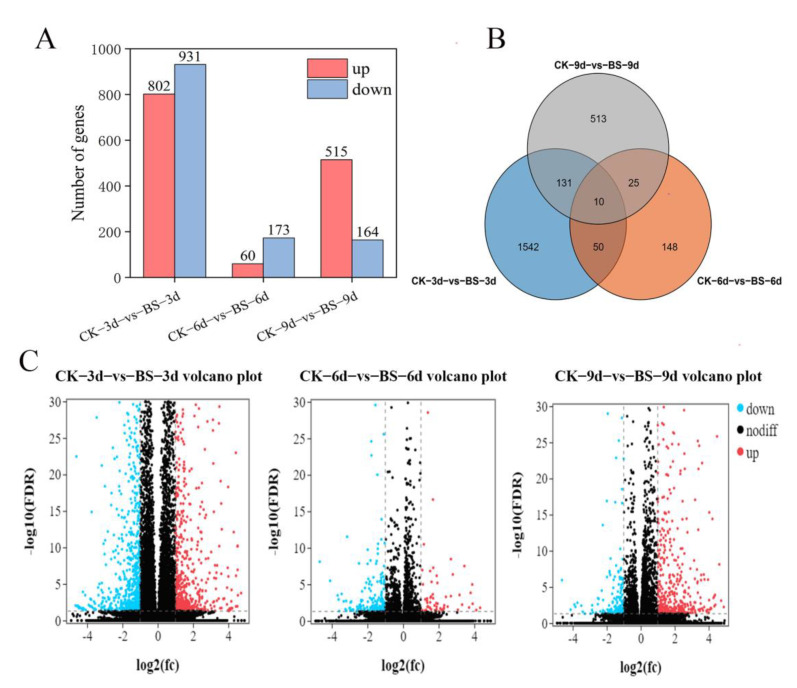
Analyze the different expressed genes in the three comparison mango fruit groups. (**A**) Bar chart of the number of DEG in three groups. (**B**) Venn diagram of DEGs among three different comparisons. (**C**) Volcano plot of DEGs.

**Figure 5 foods-11-00107-f005:**
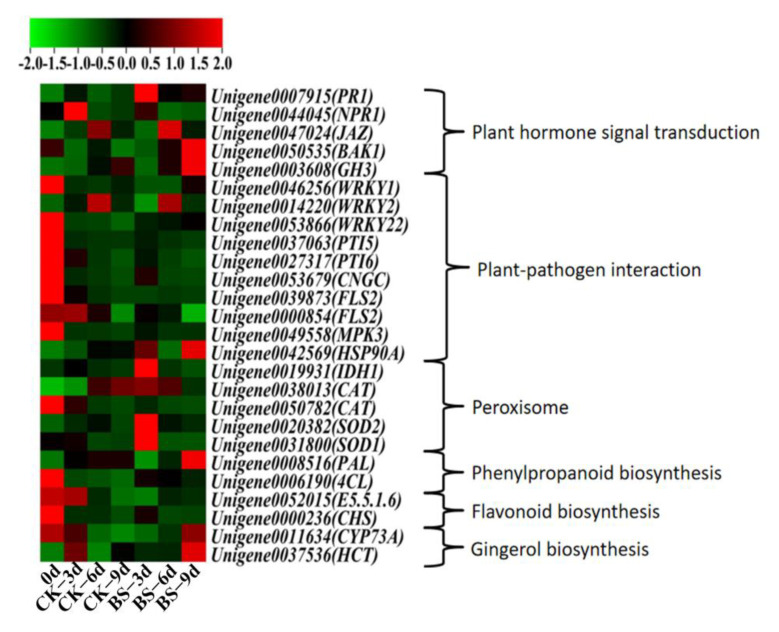
Heat map of gene expression profiles of six disease-resistant pathways that are significantly enriched in stored mango fruits.

**Figure 6 foods-11-00107-f006:**
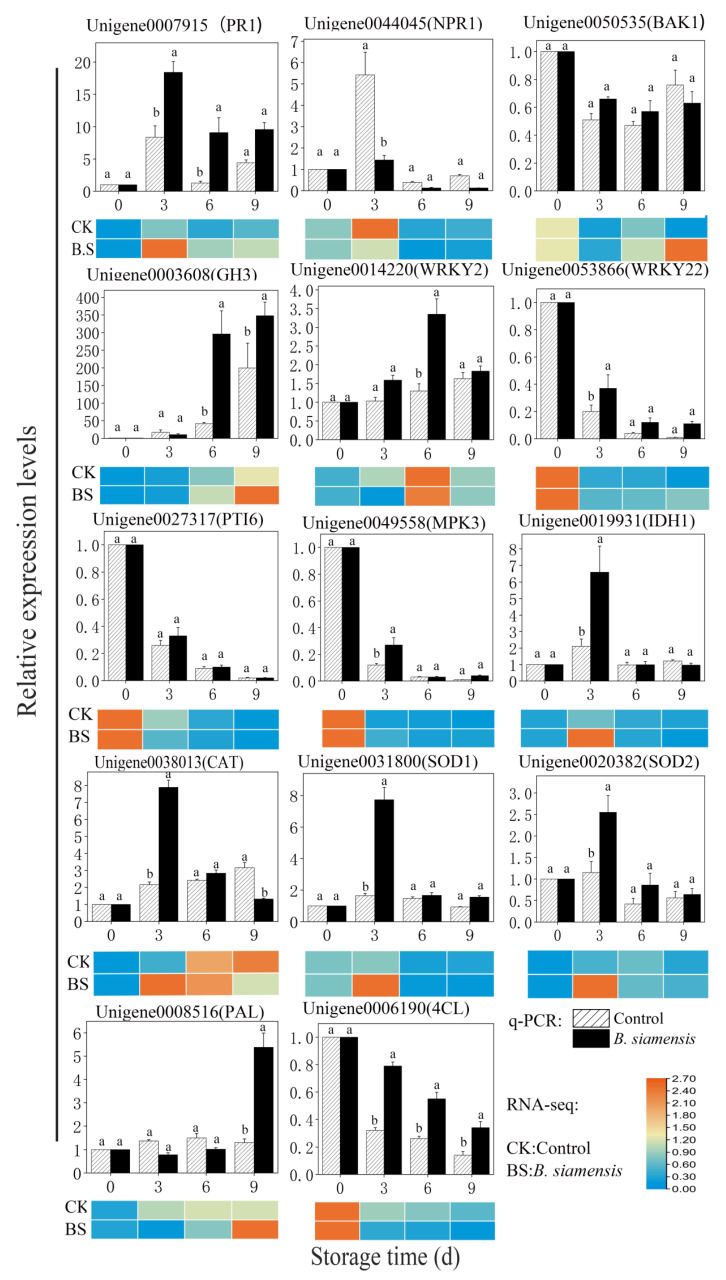
Expression pattern of 14 representative DEGs by qRT-PCR in mango fruits during storage. The expression levels of each unigenes are expressed as a ratio relative to 0 d of samples, which was set at 1. The heat map shows the change of the FPKM value of each gene. Error bars indicate standard errors of the means (*n* = 3). Bars with different lower-case letters indicate significant differences based on a t-test at the *p* ≤ 0.05 level.

**Figure 7 foods-11-00107-f007:**
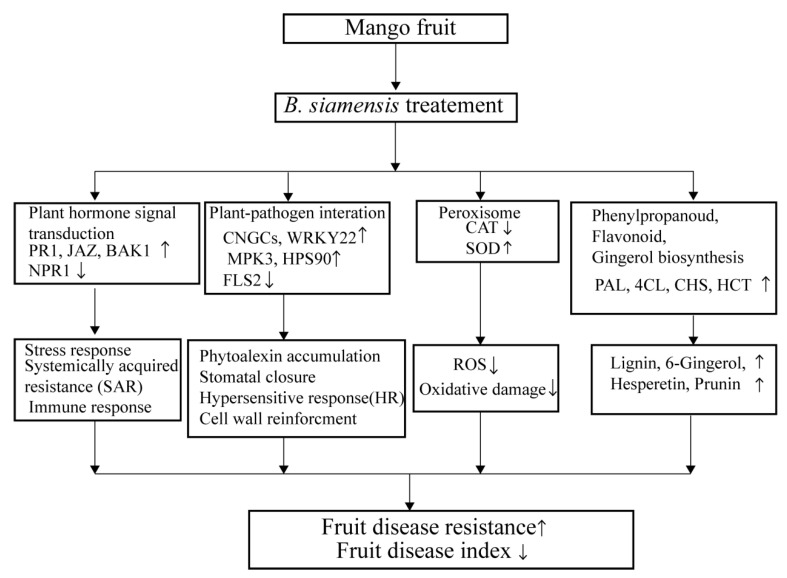
Mechanism diagram of mango defense response to *B. siamensis*. Through several branched and multi-component pathways, mangoes defense-related genes are transcribed.The up-arrow indicates the promoting effect and the down-arrow indicates the inhibiting effect.

**Table 1 foods-11-00107-t001:** Primers used in the qRT-PCR analysis.

Primers	Gene Symbol	Sequences
ACTIN-F ACTIN-R		AATGGAACTGGAATGGTCAAGGC TGCCAGATCTTCTCCATGTCATCCCA
Unigene0007915-F Unigene0007915-R	*PR1*	GCTCTCTTCTTCCCCTCCT TTCTCGCCATATTTCCCAC
Unigene0044045-F Unigene0044045-R	*NPR1*	CTCGGCCATCAGATCTCACA GAAGATCGTCACCAGCCATAG
Unigene0050535-F Unigene0050535-R	*BAK1*	CCTAACGGGAGATATTCCTGTCAATGG GAGGAGGTGGAGATACTGGAAGAGG
Unigene0003608-F Unigene0003608-R	*GH3*	ATCAGGAGGGGAGAGAAAG CACAAGGCCACCAGGAGTC
Unigene0014220-F Unigene0014220-R	*WRKY2*	CCTAACCGCCGATCAGCCATTG TCCAATCAAGAGTTCCAGCAGTAGTTC
Unigene0053866-F Unigene0053866-R	*WRKY22*	CTTCAAACAACGACAACAGCCTAAGC TTCTGGAACTTGGCAAACCCTCTTC
Unigene0027317-F Unigene0027317-R	*PTI6*	GACGACGAAGCCTGTCACCATC TGGGTTTCTTCTTTCTGGAGGGTTTG
Unigene0049558-F Unigene0049558-R	*MPK3*	ACTCTTCAGATTACACTGCCGCAATAG CTGCCTGGAAATAGAGGTCTTCTGTTC
Unigene0019931-F Unigene0019931-R	*IDH1*	CGTCATTACCGGGTTCATCAG TCCCTGATTCAACCGTTCCA
Unigene0038013-F Unigene0038013-R	*CAT*	CACATTCAAGAGGACTGGAGGATTCTG CCCAAATCATCAAACAGGAAGGTGAAC
Unigene0031800-F Unigene0031800-R	*SOD1*	ACGGCTTCCATATTCACGCTCTTG CGGCAATAATGTTACCCAAATCACCAG
Unigene0020382-F Unigene0020382-R	*SOD2*	CATCACCAGAAGCACCACCAGAC CAGACCTCCGCCGTTGAACTTG
Unigene0008516-F Unigene0008516-R	*PAL*	CTCCGTCAAGAACTGCGTCACC GGTCGTCGGCATAGCTGAACAC
Unigene0006190-F Unigene0006190-R	*4CL*	AAGAGGACGAGAGCCAA AGCCGCCCCAGATAATA

**Table 2 foods-11-00107-t002:** Summary of transcriptome data for mango fruit as well as detailed bioinformatics annotations and analyses.

**Raw Sequences and Assembly Statistics**	Number
Raw reads (paired-end)	850,798,964
clean reads (paired-end)	839,584,440
GC content percentage	38.9311
Total unigenes(average length; N50; min–max length)	56,704 (1114; 2058; 201–17,696)
**Bioinformatics Annotations of Mango Fruit Unigenes**	
Gene annotation against Nr (%)	35,499 (62.60)
Gene annotation against Swiss-Prot (%)	24,644 (43.46)
Gene annotation against KOG (%)	20,152 (35.54)
Gene annotation against KEGG (%)	33,217 (58.58)
All unigenes annotated (%)	35,648 (62.87)

## Data Availability

Data sharing not applicable.

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
