# Peer review of "Transcriptome Analysis Reveals the Inducing Effect of Bacillus siamensis on Disease Resistance in Postharvest Mango Fruit"

_foods, 2022, doi:10.3390/foods11010107_

Round 1

Reviewer 1 Report

In this article, the authors investigated the application of Bacillus siamensis on disease resistance in postharvest mango fruits. The article was well structured and well written. I would like to congratulate the authors for producing such an interesting article. findings from this article will be useful for the breeders, as well as for the post-harvest processing industry.

Author Response

Dear editor and reviewers: how are you!

Please consider the resubmitted manuscript, ‘Transcriptome Analysis Reveals the Inducing Effect of Bacillus Siamensis on Disease Resistance in Postharvest Mango Fruit ’, for publication in Foods.

This manuscript is a resubmission of Foods-1441308.

In the present version, we have decreased the repetition rate according to the iThenticate report attached. We have revised the manuscript accordingly and drafted a response to all comments from the reviewers“point by point”. All changes in the revised manuscript have been marked using the “Track Changes” function and are listed in attachment according to the editor’s and reviewers’ comments.  

We are happy to answer further questions and look forward to your decision.

Yuanzhi Shao is the corresponding author of the article, Zecheng Jiang is the first author of the article. Please communicate with us at the following e-mail address.

Yuanzhi Shao

E-mail: s.yz123789@163.com

Zecheng Jiang

E-mail: jiangzecheng516@163.com

Best wish for you and your family!

Yours sincerely!

Yuanzhi Shao, Hainan University

Reviewer 2 Report

ABSTRACT:

L13: indicate causative agent of the disease.

L13-14: further develop the part of B. siamensis. Nothing introductory is said.

KEYWORDS:

"bacillus siamensis yeast": What is that?

INTrODUCTION:

L32: Mangifera indica in italics.

L34-36: indicate causative agent of the disease.

L36-43: comment on other control strategies against postharvest diseases.

L44-45: add reference.

L44-45: Before talking specifically about B. siamensis, I think it is important to introduce the genus Bacillus.

MATERIALS AND METHODS:

Section 2.2: Does not apply a pathogen?

Section 2.3: Why not do more analysis? For example: tissue viability, ROS quantification, tissue colonization of the pathogen, etc.

Pag. 4, L18: put the exponent appropriately.

RESULTS:

Section 3.1: add other plant pathology tests.

DISCUSSION:

Pag. 13, L6-8: this information should go in the Introduction (more developed).

"In present study" is repeated a lot in this section. Modify.

Author Response

(The authors gave the same response as above.)

Reviewer 3 Report

Jiang et al presented the effect of Bacillus siamensis used as a biological agent on the transcription of the genes of mango fruits. The paper is marginally appropriate for foods as it focuses only on the transcriptional changes without providing quality-related indexes. Moreover, the discussion part is mostly descriptional with no in-depth research of the physiological phenomenon and the connection to the fruit quality and storability. 

Points for improvment. 

P2 lines 6-17. I do not understand the usefulness of this passage, the plant-pathogen interactions stated here is common knowledge.

P13 line 38 the authors make assumptions about the plant hormone levels without providing the hormonal levels. The transcriptomic analysis offers only partial information about hormonal homeostasis.

P14 line 26. The authors should provide a plausible explanation about the mechanisms governing this physiological reaction.

P14 line 32 revise “phenylalanine metabolites”

P14 line 34 revise “Pal related genes”

P14 line 41 again for the regulation of the phenylpropanoid biosynthetic pathway the authors did not propose a way that their treatments influence the physiology of the fruits.

Author Response

(The authors gave the same response as above.)

Round 2

Reviewer 3 Report

After the revision, the article can be accepted for publication.

Author Response

Dear reviewer:

Thank you very much!

Merry Christmas!

 Yuanzhi Shao

Hainan University